# Molecular Factors in PAD2 (*PADI2*) and PAD4 (*PADI4*) Are Associated with Interstitial Lung Disease Susceptibility in Rheumatoid Arthritis Patients

**DOI:** 10.3390/cells12182235

**Published:** 2023-09-08

**Authors:** Karol J. Nava-Quiroz, Jorge Rojas-Serrano, Gloria Pérez-Rubio, Ivette Buendia-Roldan, Mayra Mejía, Juan Carlos Fernández-López, Pedro Rodríguez-Henríquez, Noé Ayala-Alcantar, Espiridión Ramos-Martínez, Luis Alberto López-Flores, Alma D. Del Ángel-Pablo, Ramcés Falfán-Valencia

**Affiliations:** 1HLA Laboratory, Instituto Nacional de Enfermedades Respiratorias Ismael Cosío Villegas, Tlalpan, Mexico City 14080, Mexico; krolnava@hotmail.com (K.J.N.-Q.); glofos@yahoo.com.mx (G.P.-R.);; 2Programa de Maestría y Doctorado en Ciencias Médicas Odontológicas y de la Salud, Universidad Nacional Autónoma de México (UNAM), Mexico City 04100, Mexico; 3Rheumatology Clinic, Instituto Nacional de Enfermedades Respiratorias Ismael Cosío Villegas, Tlalpan, Mexico City 14080, Mexico; 4Translational Research Laboratory on Aging and Pulmonary Fibrosis, Instituto Nacional de Enfermedades Respiratorias Ismael Cosío Villegas, Tlalpan, Mexico City 14080, Mexico; 5Diffuse Interstitial Lung Disease Clinic, Instituto Nacional de Enfermedades Respiratorias Ismael Cosío Villegas, Tlalpan, Mexico City 14080, Mexico; 6Consorcio de Genómica Computacional, Instituto Nacional de Medicina Genómica (INMEGEN), Tlalpan, Mexico City 14610, Mexico; 7Department of Rheumatology, Hospital General Dr. Manuel Gea González, Tlalpan, Mexico City 14080, Mexico; 8Banco de Sangre, Instituto Nacional de Enfermedades Respiratorias Ismael Cosío Villegas, Tlalpan, Mexico City 14080, Mexico; 9Experimental Medicine Research Unit, Facultad de Medicina, Universidad Nacional Autónoma de México, Mexico City 06720, Mexico

**Keywords:** PAD4/*PADI4*, PAD2/*PADI2*, interstitial lung disease, rheumatoid arthritis, AIM

## Abstract

Around 50% of rheumatoid arthritis (RA) patients show some extra-articular manifestation, with the lung a usually affected organ; in addition, the presence of anti-citrullinated protein antibodies (ACPA) is a common feature, which is caused by protein citrullination modifications, catalyzed by the peptidyl arginine deiminases (PAD) enzymes. We aimed to identify single nucleotide variants (SNV) in *PADI2* and *PADI4* genes (PAD2 and PAD4 proteins, respectively) associated with susceptibility to interstitial lung disease (ILD) in RA patients and the PAD2 and PAD4 levels. Material and methods: 867 subjects were included: 118 RA-ILD patients, 133 RA patients, and 616 clinically healthy subjects (CHS). Allelic discrimination was performed in eight SNVs using qPCR, four in *PADI2* and four in *PADI4*. The ELISA technique determined PAD2 and PAD4 levels in serum and bronchoalveolar lavage (BAL) samples, and the population structure was evaluated using 14 informative ancestry markers. Results: The rs1005753-GG (OR = 4.9) in *PADI2* and rs11203366-AA (OR = 3.08), rs11203367-GG (OR = 2.4) in *PADI4* are associated with genetic susceptibility to RA-ILD as well as the ACTC haplotype (OR = 2.64). In addition, the PAD4 protein is increased in RA-ILD individuals harboring the minor allele homozygous genotype in *PADI4* SNVs. Moreover, rs1748033 in *PADI4*, rs2057094, and rs2076615 in *PADI2* are associated with RA susceptibility. In conclusion, in RA patients, single nucleotide variants in PADI4 and PADI2 are associated with ILD susceptibility. The rs1748033 in *PADI4* and two different SNVs in *PADI2* are associated with RA development but not ILD. PAD4 serum levels are increased in RA-ILD patients.

## 1. Introduction

Rheumatoid arthritis (RA) is a systemic, autoimmune, inflammatory, and progressive disease; about 50% of patients experienced some extra-articular manifestation, with lungs being a frequent site of involvement; this condition occurs in up to 67% of patients with RA, although only 10% develop a clinically diagnosed diffuse interstitial lung disease (ILD) [1,2].

In RA, specific autoantibodies have been detected as anti-citrullinated protein/peptide antibodies (ACPA), and tobacco smoking has been associated with autoimmunity and increased ACPA production [3,4]. Citrullination is a post-translational change catalyzed by peptidyl arginine deiminase (PAD) enzymes [5], modifying protein structure and increasing ACPA recognition. In the early stages of RA and idiopathic pulmonary fibrosis, soluble autoantibodies such as IgA-ACPA and soluble IgM-ACPA have been recognized, demonstrating a correlation with disease activity and smoking [6,7,8].

Members of the PAD family, PAD2 and PAD4, play roles in both synovial joint and lung tissues. Elevated levels of these enzymes and mRNA expression have been detected during inflammatory and fibrosis processes [9,10].

It has been reported that citrullinated proteins are increased in smokers’ bronchoalveolar lavage (BAL) compared to non-smokers and, therefore, associated with PAD2 and PAD4 levels in bronchial mucosa and smokers’ biopsies [11,12]. The antibodies produced against different targets (e.g., filaggrin, vimentin, and collagen), which are citrullinated, develop an immune response that triggers a perpetual inflammatory process, typical in autoimmune, chronic diseases, and pulmonary fibrosis [10,13].

Interstitial lung disease clusters many subacute and chronic respiratory conditions characterized by compromising the pulmonary parenchyma, mainly affecting the interstitium and alveolar spaces [14,15]. The lung interstitium in ILDs is involved during inflammation and fibrosis processes. In addition, patients with interstitial lung disease associated with collagen-vascular diseases have a better prognosis than people with idiopathic interstitial pneumonia [16,17,18].

In this context, PAD2 and PAD4 coding genes (*PADI2* and *PADI4*, respectively) have single nucleotide variants (SNV), which could lead to a critical role in the folding, activity, function, or half-life of PAD proteins [19,20]. We aimed to identify the relationship between polymorphisms in *PADI2* and *PADI4* in developing interstitial lung disease in patients with rheumatoid arthritis, their correlation with the PAD2, PAD4 protein levels, and clinical characteristics in RA-ILD.

## 2. Materials and Methods

### 2.1. Subjects, Material, and Methods

The cross-sectional study included three groups: an RA-ILD patients’ group, patients with RA without ILD, and a clinically healthy subjects’ group. Genetic association analyses were performed. In addition, we determined the PAD2 and PAD4 protein levels in a subgroup of subjects.

### 2.2. Ethical Statement

The Institutional Committees for Research, Biosecurity, and Ethics in Research of the Instituto Nacional de Enfermedades Respiratorias Ismael Cosío Villegas (INER) approved this study (approbation codes B20-15 and C08-15). All participants authorized and signed the corresponding informed consent. The Instituto Nacional de Enfermedades Respiratorias Ismael Cosio Villegas (INER) provided a document guaranteeing that personal data were protected, incorporated, and processed as sensitive and personal data.

### 2.3. Study Groups

The ILD patients were recruited from the Interstitial Lung Disease and Rheumatology Unit (ILD&RU) at INER. The RA without ILD patients were recruited from the Department of Rheumatology of the Hospital General Dr. Manuel Gea González, Mexico City, Mexico.

Pneumologists and rheumatologists evaluated the patients. For the RA-ILD and RA groups, pulmonary function tests (PFTs), chest radiography, and tomography were performed in the Translational Research Laboratory on Aging and Pulmonary Fibrosis at INER and the Department of Respiratory Physiology of INER. In each measurement of PFTs, weight, and standing height were measured with a digital scale (models 206 and 769, Seca, Hamburg, Germany).

Spirometry (to obtain forced vital capacity) and diffusing capacity for carbon monoxide test (DLCO) were performed using Easy One Pro Lab (Ndd Zurich, Zürich, Switzerland). The data were expressed as percentages of the predicted values; each subject’s predicted values were obtained according to PLATINO and NHANES studies [21,22]. All spirometry tests fulfilled the acceptability and reproducibility criteria (ATS/ERS) [23,24,25]. All patients present treatment with immunosuppressive therapy: methotrexate, prednisone, chloroquine, or hydroxychloroquine, among others (or combination). None of the patients were undergoing antifibrotic therapy at the time of sample collection.

The third comparison group included clinically healthy subjects (CHS), volunteers enrolled from the biobank of the HLA Laboratory, and the INER’s blood donors bank, all of them without pulmonary or rheumatic diseases, including smokers (current and former) selected according to age, sex, and tobacco index regarding the patients’ groups.

All included individuals self-referred as Mexican Mestizos with three generations born in México and not biologically related.

### 2.4. HRCT Scan Pattern

High-resolution computed tomography was performed on patients with RA-ILD, interpreted by medical personnel specializing in interstitial diseases, determining the tomographic pattern [26].

### 2.5. Biological Samples Collection

Blood samples were drawn by puncturing the forearm vein using the BD Vacutainer vacuum system in tubes with EDTA as an anticoagulant (Sarstedt S-Monovette 7.5 mL, EDTA KE, Nümbrecht, Germany) to obtain DNA and a tube with a clot activating gel to obtain serum (Sarstedt S-Monovette 7.5 mL, Serum-Gel, Nümbrecht, Germany).

The collection of 10 mL of bronchioalveolar lavage (BAL) fluid from 22 patients with RA-ILD was carried out. A specialist in bronchoscopy performed the procedure, and only in those patients who required it as a diagnostic test and for discarding other lung diseases; the samples were stored at −80 °C until analysis.

DNA was obtained using the BDtrack Genomic DNA Isolation Kit (Maxim Biotech, CA, USA). The DNA purification procedure was carried out according to the manufacturer’s instructions. DNA hydration was carried out with 250 µL of TE buffer (Ambion, Waltham, MA, USA). Samples were stored at 4 °C. The DNA extracted was quantified using the Nanodrop 2000 device (Thermo Scientific, Wilmington, DE, USA).

### 2.6. Genotyping

Allele discrimination assay was performed on eight SNV in *PADI2* and *PADI4* by real-time PCR, using TaqMan probes (Applied Biosystems, Foster City, CA, USA), *PADI2* (rs2235926, rs2057094, rs2076615, and rs1005753), and *PADI4* (rs11203366, rs11203367, rs874881, and rs1748033), which have previously been identified as associated with susceptibility to RA; however, they have not been described in extraarticular manifestations, such as the lungs.

The DNA was adjusted to a final concentration of 15 ng/µL. Then, a reaction mixture was prepared with the TaqMan probe and TaqMan Master Mix™ (Applied Biosystems, CA, USA) and nuclease-free water. It was mixed and centrifuged at 1500 rpm, then entered into the 7300 Real-Time PCR system (Applied Biosystems, CA, USA) in the thermal cycler according to the conditions indicated by the manufacturer.

### 2.7. Ancestry Informative Markers

Additionally, a panel of 14 SNVs was considered to analyze ancestry-informative markers (AIMs) in the whole population (patients and all control subjects). The SNVs were selected according to their frequency in the Mexican Mestizo population [27], with two reference populations: Utah Residents with Northern and Western European ancestry (CEU) as Caucasian and Zapotecs from the state of Oaxaca, Mexico (ZAP) as Amerindians of the Phase 31,000 Genomes project and International HapMap Project (phase 3) (websites: http://browser.1000genomes.org, accessed: 1 July 2021 and https://ftp.ncbi.nlm.nih.gov/hapmap, accessed: 1 July 2021, respectively). The results obtained from AIMs (eigenvalues 1 and 2) were included as covariates in the logistic regression model.

### 2.8. Determination of PAD2 and PAD4 Proteins

PAD2 and PAD4 quantifications from serum and BAL were performed using the enzyme-linked immunosorbent assay (ELISA) technique using PAD2 and PAD4 human ELISA kits (part number: 501450 and 501460, Cayman Chemical, Ann Arbor, MI, USA), following manufacturer’s instructions. The detection kit range is 0.156–10 ng/mL, and we used 1:100 and 1:1000 dilutions in samples that exceeded the detection range. Then, 4 mL of BAL supernatant from patients with RA-ILD were concentrated in the Vacufuge Concentrator (Eppendorf, Framingham, MA, USA) at 14,000 rpm for 12 h, then resuspended in 200 µL of immunoassay buffer, and the assay was carried out according to protocol. Determining PAD2 and PAD4 proteins in each patient was performed by duplicating; the reading was performed in the iMark™ Microplate Absorbance Reader (Bio-Rad, Hercules, CA, USA) at a wavelength of 450 nm.

### 2.9. Statistical Analysis

The Kolmogorov–Smirnov data normality test was performed. Demographic and clinical variables in RA and RA-ILD patients were evaluated through the exact Fisher test in categorical variables, U Mann–Whitney in continuous quantitative variables, and a Spearman “ρ–rho” correlation analysis. For the comparisons of protein concentrations according to the SNVs’ genotypes, we used the Kruskal–Wallis test, with later correction with the Bonferroni test, in software R v3.5.1 [28].

The analysis of allele and genotype frequencies was performed through PLINK v1.07 software [29] using Fisher’s exact test, taking a value of *p* < 0.05 as significant, Odds Ratio (OR), and 95% confidence intervals (95% CI). Haplotype analysis was carried out with Haploview v4.2 [30]. The principal components analysis (PCA) in the EIGENSTRAT v3.0 software, using the genotyping of 14 SNVs AIMs previously described [31,32].

A logistic regression model was performed, adjusted by age, sex, tobacco index, and AIMs (Eigenvalue1 and Eigenvalue2 obtained of PCA), which provided the maximum variance [33] in PLINK v1.07.

## 3. Results

### 3.1. Demographic and Clinical Data

Table 1 shows the clinical and demographic characteristics of the patients’ groups and the group of clinically healthy subjects; 118 rheumatoid arthritis patients with interstitial lung disease (RA-ILD) and 133 patients with rheumatoid arthritis (RA) were included (Figure 1) with the risk factors, such as tobacco smoking, exposure to birds, biomass-burning smoke, and other exposures (use of oil or petroleum, and exposure to occupational factors).

In addition, 616 clinically healthy subjects were included; the median age was 49.24 years, 58% were men, and 42.19% were current and former smokers. Spirometry parameters indicate normal pulmonary function and exposure factors, as shown in Table 1.

In the biochemical data regarding anti-CCP antibodies, Table 1 shows that RA-ILD patients have lower anti-CCP levels (150.49 IU/mL) than the RA group (261 IU/mL). There is no difference in the levels of rheumatoid factor, C-Reactive protein, or the erythrocyte sedimentation rate between the groups of patients with RA-ILD vs. RA.

There are differences between the groups in the spirometry parameters because the RA-ILD patients present a restriction pattern: median FVC (%) = 66 vs. 94.5 in the group of RA patients without lung disease.

Around 25% of RA-ILD patients were tobacco smokers, with a higher consumption rate vs. 15% in the RA group, in addition to differences in exposure to biomass-burning smoke and exposure to birds, with a higher frequency in the RA-ILD group. It should be mentioned that in the group of patients with RA, about 74% of the patients do not present an exposure to environmental factors as well as 24% in the RA-ILD group.

### 3.2. Genetic Susceptibility Analysis

Hardy Weinberg equilibrium (HWE) was calculated on all SNVs in both genes. The rs2235926 in *PADI2* does not comply with HWE (*p* = 8.9 × 10^−24^); consequently, this SNV was not considered for further analyses.

The AIM contribution was calculated using the 14 SNVs in the principal components analysis, shown in the Appendix A.

The fixation index (FST) was evaluated between groups RA-ILD vs. RA, RA-ILD vs. CHS, and RA vs. CHS; we found a statistically significant difference between the RA and CHS groups (*p* = 0.043) (Appendix A). Because of this difference, we use PCA eigenvectors 1 and 2 in the logistic regression model.

### 3.3. Comparison of Rheumatoid Arthritis and Clinically Healthy Subjects

To evaluate genetic association with rheumatoid arthritis (Arthritis^++^), all patients having RA, including those with ILD (from the RA-ILD group), were compared to the CHS group.

Of the three evaluated SNVs in *PADI2* (Table 2), the rs2057094-AA homozygous genotype was associated with a risk (OR = 1.71, 95% CI = 1.19–2.44, *p* = 0.015); homozygous genotypes to the minor allele rs2076615-CC and rs1005753-GG showed a decreased risk of RA (OR = 0.47 and 0.40, respectively, *p* < 0.05). The same behavior was observed for the alleles. The rs2057094-A allele is associated with an increased risk of RA (OR = 1.45, 95% CI = 1.17–1.80, *p* = 0.003), while rs2076615-C and rs1005753-G were associated with a decreased risk of RA (OR = 0.71 in both alleles, *p* < 0.05).

Of four SNVs evaluated in *PADI4* (Table 3), the homozygous of rs1748033-TT was associated with an increased risk (OR = 2.26, 95% CI = 1.43–3.56, *p* = 0.002), and in rs874881- GG associated with a lower risk (OR = 0.43, 95% CI = 0.27–0.68, *p* = 0.001]). Similar to the previously observed in the analysis of the genotypes, the minor allele of rs1748033-T was associated with risk (OR = 1.47 [1.18–1.83], *p* = 0.002), and the rs874881-G was associated with a decreased risk (OR = 0.69 [0.56–0.86], *p* = 0.004).

### 3.4. Comparison of RA-ILD and RA Groups

On the patients’ groups comparisons (RA-ILD vs. RA), only an SNV in *PADI2* was found to be significantly associated; homozygous genotype rs1005753-GG shows a strong association with the risk as well (OR = 4.91, 95% CI = 1.00–24.05), and the minor allele of rs1005753-G showed an increased risk (OR = 1.71 [1.10–2.64]) (see in Table 2).

Table 3 shows genotype and allele comparisons of SNVs in *PADI4*; three are associated with an increased risk: homozygous genotype of rs11203366-AA (OR = 3.08, 95% CI = 1.40–6.74, *p* = 0.004) and the minor allele rs11203366-A (OR = 1.7, 95% CI = 1.18–2.47, *p* = 0.005); in the rs11203367, the homozygous CC genotype (OR = 2.4, 95% CI = 1.03–5.58, *p* = 0.038) and the allele rs11203367-C (OR = 1.5 [1.03–2.18, *p* = 0.037]); and finally, the genotype rs874881-GG (OR = 2.53, 95% CI = 1.15–5.58, *p* = 0.002) and the G allele (OR = 1.7, 95% CI = 1.18–2.45, *p* = 0.003).

### 3.5. Haplotypes

Haplotype analysis was performed, and two blocks were identified. The first in *PADI2* was created by two SNVs (rs2057094 and rs2076615) and four in *PADI4* (by the rs11203366, rs11203367, rs1748033, and rs874881). The linkage disequilibrium diagram is shown in Table 4 and Table 5, and eight haplotypes were obtained, of which four are statistically significant in comparing frequencies between individuals with RA (with and without ILD) versus the CHS group (shown in Table 4 and Figure 2a). The GTTC haplotype in *PADI4* was associated with an increased risk of developing arthritis (OR = 1.40, 95% CI = 1.12–1.74, *p* = 0.018) while three haplotypes (GTCG, ACTC, and ACTG) were associated with decreased risk (OR = 0.12 to 0.65, [*p* < 0.05]). Furthermore, seven haplotypes in *PADI4* were found in the RA-ILD vs. RA comparison; the ACTC haplotype is associated with ILD susceptibility in RA patients (OR = 2.64, 95% CI = 1.01–6.88, *p* = 0.038), which has a high linkage disequilibrium (r^2^ value ≥ 80), shown in Table 5 and Figure 2b.

### 3.6. PAD2 and PAD4 Protein Levels

The protein quantification was performed in patients with available serum and BAL samples; in addition, we evaluated serum protein levels in a subgroup of CHS. Appendix A shows the demographic and clinical variables for evaluated subjects. Serum PAD2 levels in individuals with RA-ILD were ~3 ng/mL, 2.85 ng/mL in healthy individuals, and 4.87 ng/mL in the RA group. In the BAL of RA-ILD patients, the detected PAD2 levels were 0.3 ng/mL. The median concentration of PAD4 serum levels in RA-ILD is 8.65 ng/mL, 7.91 ng/mL in the RA group and 1.05 ng/mL in CHS. The BAL levels were ~10 ng/mL in the RA-ILD group. The Kruskal–Wallis test was performed in the groups’ comparisons; no significant difference was identified in the serum PAD2 determination. There is no difference in the PAD4 serum levels in the RA and RA-ILD comparison; however, in the RA vs. CHS, increased levels in the RA group were found (*p* < 0.001). In addition, differences were observed in the RA-ILD group vs. CHS (*p* = 0.001).

### 3.7. PAD2 and PAD4 Levels in the Codominant Genetic Model

The comparison of the levels of PAD2 and PAD4 proteins and genotypes in a codominant genetic model was carried out among the three study groups. Statistically significant genotypes related to PAD2 and PAD4 serum levels are observed.

For the PAD2/*PADI2* analysis, the rs2076615-AC and rs1005753-TG heterozygous genotypes are associated with decreased PAD2 levels in the RA-ILD group vs. RA group (*p* < 0.05), the figures are shown in the Appendix A and serum levels depending on genotypes are included in the Appendix A.

For the PAD4/*PADI4* analysis, the homozygous genotypes to the minor allele in the rs11203367-CC and rs1748033-TT SNVs (see Figure 3a,b) were found to be associated with PAD4 increased levels in the RA-ILD patients comparing to the RA group. Contrarily, in the two remaining variants in *PADI4*, the homozygous allele to the common allele showed an association with increased levels in the same comparison, RA-ILD vs. RA (Figure 3c,d). The information, including PAD4 levels by genotypes, can be accessed in Appendix A.

### 3.8. Correlation of PAD2 and PAD4 Levels with Clinical and Biochemical Variables

PAD2 and PAD4 correlation analyses were performed in the serum and BAL of 22 patients with RA-ILD. The r values obtained from the correlations are shown in Figure 4. Clinical and biochemical data at the moment of the ILD identification were considered. The diagnosis variables of rheumatoid arthritis and interstitial lung disease were used for the correlation analysis: age, anti-CCP, rheumatoid factor (RF), C-Reactive protein (CRP), forced vital capacity (FVC), expiratory volume in the first second (FEV_1_), and the PAD2 and PAD4 protein levels in serum and bronchoalveolar lavage (RA-ILD patients).

The PAD4 serum levels (PAD4_S) correlated positively with RF in serum (ρ = 0.43). In addition, in BAL, PAD4 correlated negatively with RF (ρ = −0.59). The correlation plot shows only statistically significant *p*-values (*p* < 0.05) in color. The correlations obtained from serum PAD2 (PAD2_S) correlate negatively with FEV_1_, FVC, and age (ρ = −0.25, −0.31, and −0.36, respectively) and positively with anti-CCP (ρ = 0.43). While in BAL, PAD2 correlates with anti-CCP (ρ = 0.35), FVC (ρ = −0.33) and PAD2 serum levels (ρ = 0.36).

## 4. Discussion

Our study demonstrates the participation of PAD2 and PAD4 enzymes genetic polymorphisms in the susceptibility to develop interstitial lung disease (ILD) in individuals with rheumatoid arthritis (RA) as well as the increase in PAD4 levels in subjects carrying risk genotypes in genes that code for this enzyme.

Among participants, the ILD was more frequent in men than women. RA-ILD reaches 32.20% in men while men with only RA were 7.52%, consistent with previous reports [34], while RA is in women [35]. In addition, the age at which these diseases occur and the prognosis for each of them is variable. The age in RA-ILD patients was 61.5 years and 54 years in the RA group. In the United States of America, 68% of individuals over 65 years of age who have interstitial lung disease are men [36].

The anti-CCP levels in RA individuals were increased (261 IU/mL) compared to the RA-ILD group (150.49 IU/mL). The autoantibody generation, still a not-fully described topic, and the increased levels of autoantibodies’ anti-CCP in RA patients have been reported; likewise, increased proinflammatory cytokine levels have been informed among patients with positive autoantibodies vs. negative in individuals with RA, such as the interleukin 6 (IL-6, up to 4-fold) [37].

Recently, Arnoux et al. described an increase in ACPA response using the experimental hapten (citrullinated proteins)/carrier (PAD) model. However, its production mechanism is unclear; some authors suggest that *HLA-DR* alleles bind citrullinated peptides in those involved in PAD2 and 4 (upregulated by tobacco smoking), which are subsequently recognized by T cells, producing ACPAs, which are detected like anti-CCP in serum [38,39] and provide an enhancer stimulus to the effect of these pathologies.

According to environmental risk factors, tobacco smoking was the most common among RA-ILD patients (~25%). In comparison, the RA group just reached 15%, probably due to more women in the RA group. According to the National Health Survey (ENSANUT), most of the tobacco smokers in Mexico are men [40]. With exposure to biomass-burning smoke, 26% of individuals had RA-ILD, and 15% had RA.

Tobacco smoking is the most associated factor in the susceptibility and development of pulmonary and autoimmune diseases. However, up to 60% of individuals with interstitial lung disease are never-smokers, similar to what is observed in this study where about ~25% in RA-ILD and 74% in RA do not present this exposure, which suggests the existence of other exposures to environmental or genetic factors participating in the development of these diseases [36].

In developing countries such as Mexico, biomass burning and its exposure to smoke derived from its combustion are other environmental risk factors rarely considered in developed countries’ studies. However, this exposure has focused on other lung diseases, such as asthma and COPD [41,42,43], ignoring its role in RA-ILD. As observed in this study, exposure to biomass burning is the second environmental risk factor in ILD.

In serum determination, PAD2 proteins in the RA group were increased compared to the clinically healthy and RA-ILD groups, but in the BAL sample, they were very low (0.3 ng/mL). Previously, Badillo Soto and colleagues identified that the expression of PAD2 was exclusive in patients with RA; however, the sample used in their analysis was synovial biopsies from RA patients [44], and in the present study, serum samples from both RA and RA-ILD patients were analyzed, along with those in BAL samples of RA-ILD patients.

PAD4 serum levels were increased in the RA-ILD group (8.65 ng/mL) compared to clinically healthy subjects. In the RA group, the levels were similar to the RA-ILD group (7.91 ng/mL). Interestingly, in BAL, the levels were slightly high (10 ng/mL); however, there is no correlation between the serum and BAL of RA-ILD patients as opposed to quantification with BAL PAD2 levels that were undetectable [45,46]. The presence of autoantibodies (anti-PAD4 and anti-CCP) was observed in a follow-up from a pre-clinical period, which could increase over time [47]. Moreover, it is associated with the disease’s severity or activity and could bias this study.

In genetic association studies, the variability concerning populations has been noticed; for this reason, AIMs were employed. Silva-Zolezzi reported 14 SNVs that show a geographical and genetic variation between the population of México, which provide high informative content and greater genetic diversity intrapopulation [27]. Therefore, AIMs were included as covariates of fit in genetic analysis.

In our study, the rs1005753-GG is associated with a greater ILD risk in RA individuals. The rs2057094-AA is associated with an RA risk, but the rs2076615-CC has a lower RA risk. In the Asian population of *PADI2*, the rs1005753 found the T allele associated with a lower risk of developing RA, contrary to our findings. They also showed that rs2057094-A, with an increase in RA’s susceptibility, was similar to our results [48,49].

In *PADI4*, the genotypes rs11203366-AA, rs11203367-CC, and rs874881-GG were associated with a susceptibility to interstitial lung disease in individuals with rheumatoid arthritis and rs1748033-T with a susceptibility to RA in comparison with CHS subjects. The rs874881 was associated with decreased risk (or minor susceptibility) to RA and increased risk to RA-ILD because of the low frequency of ILD in RA patients. In the haplotypes’ analysis, the GTTC haplotype was associated with RA susceptibility while ACTC was associated with ILD risk in individuals with RA.

Polymorphisms occur in *PADI4*, which are associated with a susceptibility to RA, such as in our findings, and interactions between homozygous to the GTG haplotype (rs11203366, rs11203367, and rs874881) of SNV in *PADI4* and the *HLA-DRB1* shared epitope (SE) allele, which are related to the anti-CCP production, as well as smoking and the presence of erosive disease of the same patients [50]. This haplotype has been described as susceptible to developing RA in smoking individuals with positive ACPA and HLA alleles with SE [50,51]. In addition to the risk each SNV gives individually, it has a high linkage disequilibrium, described in different populations.

Different *PADI4* polymorphisms and haplotypes (same SNVs) have been described and replicated in diverse populations. Indian populations [52] are associated with a lower risk of RA (GTGC and ACCC haplotypes), and the haplotype GTG is associated with an increased risk of RA in Mexican [53], French [54], and Korean [19,50] populations, where they have been associated with RA. However, Swedish [55] and Chinese [56,57,58] populations were not statistically significant for RA. In the Japanese population, they observed the predisposition of men who are smokers to developing rheumatoid arthritis [59]. However, the haplotype ACTC found in this study had not been described in ILD susceptibility in RA patients.

Protein levels were analyzed according to the risk genotype that individuals carried. Taking into account the PAD4 proteins, the SNVs in PAD that encode them, and the risk factors (environmental exposures) of individuals with the diseases, PAD4 levels increased in individuals with RA-ILD that are homozygous at a minor allele associated with a risk of RA-ILD, the rs11203367-CC, rs1748033-TT, and increased in those homozygous at the common allele rs11203366-GG, rs874881-CC. In the first SNV, the risk increased in PAD4 levels in individuals carrying this SNV and the ILD compared with RA.

When comparing SNVs in *PADI2* with PAD2 serum levels, the heterozygous genotype is associated with an increase in levels of RA patients in rs1005753-TG and rs2076615-AC, not reported until now; however, this could be due to the low frequency of the minor allele homozygous in the Mexican population.

Citrullination at different sites may explain an inactivation of PAD proteins. The citrullination process may modify the affinity, interaction, and media life. Other authors have suggested that auto-citrullination regulates the production of citrullinated proteins during cell activation, and this was affected by polymorphisms in *PADI* genes, both its structure and the immune response [60,61]. Therefore, PAD4/*PADI4* could be a marker in interstitial lung diseases.

One of the study’s limitations was that some patients have treatment for both RA and RA-ILD, and the patients’ recruitment was carried out in a pulmonology specialty institute; of this, the sample size of patients with RA was smaller.

## 5. Conclusions

The polymorphisms rs11203366; rs11203367, rs874881 in *PADI4*; and rs1005753 in *PADI2* are associated with developing ILD in patients with RA. The SNVs rs1748033 in *PADI4*, rs2057094, and rs2076615 in *PADI2* were associated with AR development but not with ILD as well as haplotype in *PADI4* ACTC in patients with RA-ILD. Finally, the levels of PAD4 protein are increased in ILD patients.

## Figures and Tables

**Figure 1 cells-12-02235-f001:**
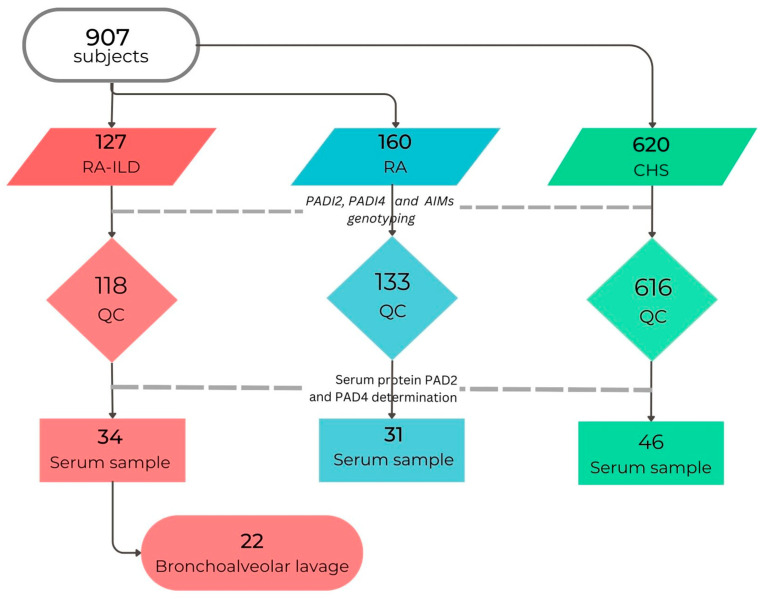
Subjects included in the study. QC: Quality Control, patients with asthma, and subjects with <95% genotyped SNVs were excluded. RA: rheumatoid arthritis, RA-ILD: interstitial lung disease associated with RA, CHS: clinically healthy subjects, BAL: bronchoalveolar lavage.

**Figure 2 cells-12-02235-f002:**
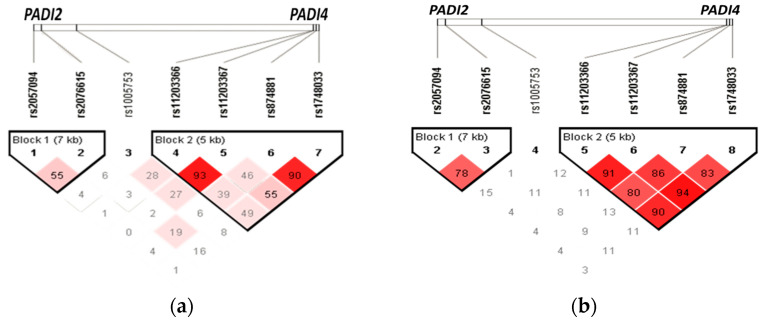
Haplotypes in *PADI2* and *PADI4.* (**a**) All Rheumatoid Arthritis patients vs. CHS; (**b**) RA-ILD vs. RA patients. Linkage disequilibrium (LD) is shown in red, with higher intensity indicating a stronger LD.

**Figure 3 cells-12-02235-f003:**
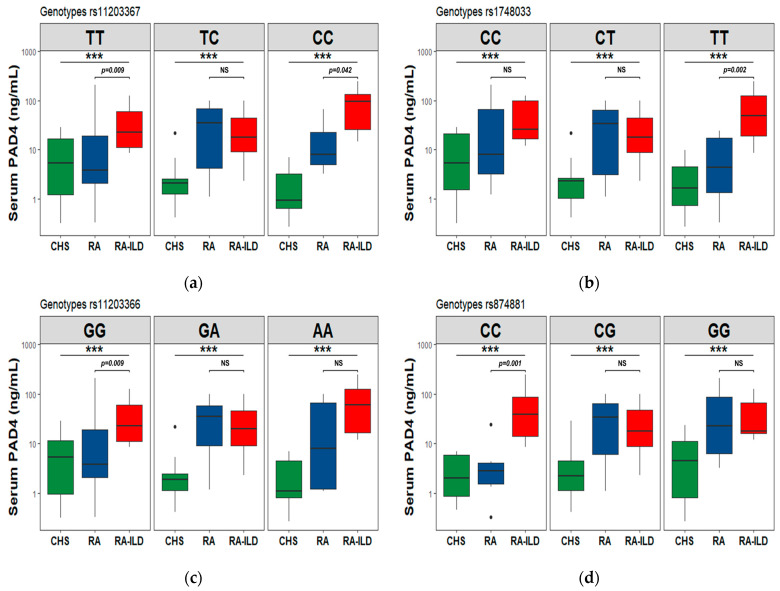
Associations among serum PAD4 levels and *PADI4* SNV genotypes between study groups. The analysis was performed using the Kruskal–Wallis test (***) and corrections for multiple tests using the Bonferroni method. Show *p*-value (Bonferroni post hoc) only in comparison to RA-ILD vs. RA groups. CHS: Clinical healthy subjects, RA-ILD: Interstitial lung disease–Rheumatoid Arthritis patients’ group. (**a**) serum PAD4 levels between rs11203367 genotypes; (**b**) rs1748033; (**c**) rs11203366; (**d**) rs874881. The points outside the boxes and whiskers are considered outliers.

**Figure 4 cells-12-02235-f004:**
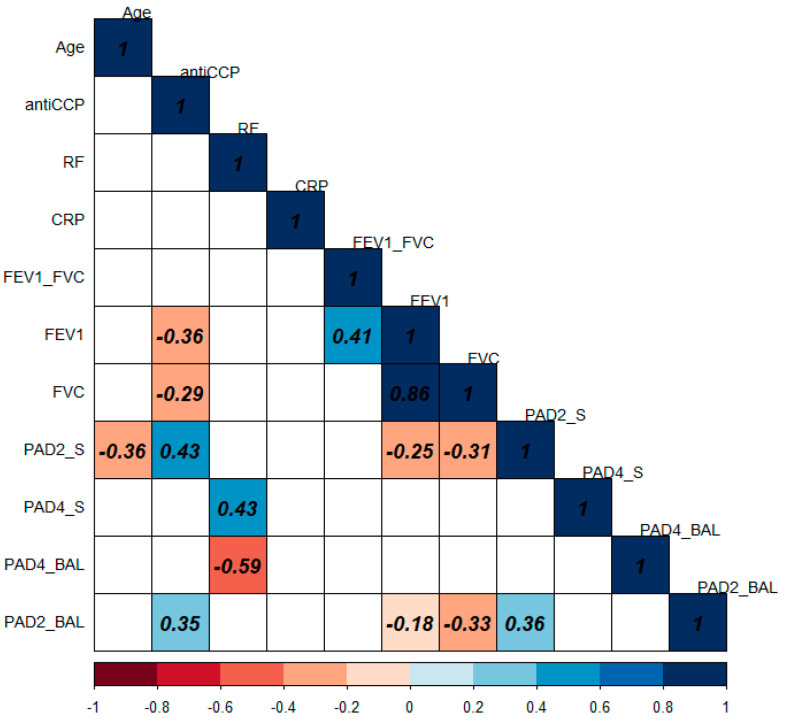
Correlation plot of 22 RA-ILD patients between protein PAD2 and PAD4 levels and clinical variables. Red or blue shows only the significant (*p* < 0.05) Spearman correlations.

**Table 1 cells-12-02235-t001:** Demographic and clinical characteristics of the patients’ groups.

	RA-ILD (*n* = 118)	RA (*n* = 133)	CHS (*n* = 616)	*p*-Value *
Male, *n* (%)	38 (32.20)	10 (7.52)	358 (58%)	<0.01
Age (years)	61.50 (53–67)	54 (44–62)	49.24 (42–56)	<0.01
RA onset (years)	52 (42–59)	48 (38–59)	NA	<0.01
Biochemical data				
anti-CCP (IU/mL)	150.49 (70.93–201)	261 (130–345.6)	NA	<0.01
Rheumatoid Factor (IU/mL)	322 (64.17–761)	182 (55.15–656)	NA	0.153
C-Reactive protein (mg/dL)	1.65 (0.52–3.61)	1.39 (0.40–3.30)	NA	0.625
Erythrocyte sedimentation rate (mm/h)	33 (24–36)	29 (19.50–43)	NA	0.891
Lung Function				
FEV_1_ (%)	55.50 (26–89)	75.50 (64–99)	99 (88–108)	<0.01
FVC (%)	66 (27–98)	94.50 (75.5–113)	97 (88–10.50)	<0.01
FEV_1_/FVC (%)	81.50 (30–103)	83 (58–110)	81 (77–85.60)	0.04
Exposure factors, *n* (%)				
Tobacco smoking (%)	30 (25.42)	20 (15.04)	260 (42.19)	0.04
Tobacco index	7.50 (2.40–20)	2 (1.43–5.38)	6 (3.20–9.0)	0.02
Biomass-burning smoke	31 (26.27)	20 (15.04)	65 (10.55)	0.01
Birds	21 (17.79)	7 (5.26)	37 (6.01)	<0.01
Others	16 (13.56)	14 (10.53)	68 (11.04)	0.33
No exposure	29 (24.57)	99 (74.44)	319 (51.79)	<0.01

Quantitative variables are expressed in median (interquartile range) and categorical in number (percentage). RA-ILD: Rheumatoid Arthritis–Interstitial Lung Disease-; anti-CCP: anti-cyclic citrullinated peptide; FEV1: forced expiratory volume in the 1st second; FVC: forced vital capacity, and ratio FEV_1_/FVC. * *p*-value in comparison of patient groups RA-ILD vs. RA.

**Table 2 cells-12-02235-t002:** Analysis of genotypes and alleles of SNV in *PADI2*.

*PADI2* *SNV*	RA-ILD (*n* = 118)	RA (*n* = 133)	CHS (*n* = 616)	Arthritis^++^ vs. CHS			RA-ILD vs. RA		
	F%	F%	F%	*p*-Value	OR	95% CI	*p*-Value *	OR	95% CI
rs2057094									
GG	35.59	40.54	43.67	0.015	1		0.612		
GA	23.73	22.52	30.19		0.88	0.60–1.30			
AA	40.68	36.94	26.14		1.71	1.19–2.44			
G	47.46	51.8	58.77	0.003	0.69	0.56–0.86	0.400		
A	52.54	48.2	41.23		1.45	1.17–1.80			
rs2076615									
AA	46.09	42.73	30.86	0.001	1		0.920		
AC	50.43	53.64	63.86		0.57	0.41–0.78			
CC	03.48	03.64	05.28		0.47	0.21–1.05			
A	71.30	69.55	62.79	0.012	1.41	1.11–1.78	0.757		
C	28.70	30.45	37.21		0.71	0.56–0.89			
rs1005753									
TT	48.72	63.06	46.72	0.015	1		0.041	1	
TG	44.44	35.14	44.10		0.76	0.55–1.04		1.63	0.95–2.82
GG	06.84	01.80	09.18		0.40	0.19–0.81		4.91	1.00–24.05
T	70.94	80.63	68.77	0.030	1.41	1.10–1.80	0.017	0.59	0.38–0.90
G	29.06	19.37	31.23		0.71	0.55–0.91		1.71	1.10–2.64

F%: Frequency in percentage; ^++^ All individuals with RA and RA-ILD are included compared to individuals without diseases CHS. *p* * = adjusted for age, sex, tobacco index, and AIMs panel. A value of *p* < 0.05 is considered significant, OR: Odds ratio, CI: 95% confidence interval.

**Table 3 cells-12-02235-t003:** Analysis of genotypes and alleles of SNV in *PADI4*.

*PADI4* *SNV*	RA-ILD (*n* = 118)	RA (*n* = 133)	CHS (*n* = 616)	Arthritis^++^ vs. CHS			RA-ILD vs. RA		
	F%	F%	F%	*p*-Value	OR	95% CI	*p*-Value *	OR	95% CI
rs11203366									
GG	21.55	35.96	30.50	0.852			0.004	1	
GA	52.59	50.00	47.00					1.76	0.95–3.25
AA	25.86	14.04	20.90					3.08	1.40–6.74
G	47.84	60.96	54.85	0.869			0.005	0.59	0.41–0.85
A	52.16	39.04	45.14					1.70	1.18–2.47
rs11203367									
TT	22.52	36.52	29.70	0.319			0.038	1	
TC	59.46	51.30	48.30					1.88	1.02–3.45
CC	18.02	12.17	20.60					2.40	1.03–5.58
T	52.25	62.17	54.60	0.346			0.037	0.67	0.46–0.97
C	47.75	37.83	45.40					1.50	1.03–2.18
rs1748033									
CC	16.38	14.91	25.10	0.002	1		0.998		
CT	50.00	51.57	49.80		1.64	1.08–2.50			
TT	33.62	33.33	23.80		2.26	1.43–3.56			
C	41.38	40.79	50.66	0.002	0.68	0.55–0.84	0.925		
T	58.62	59.21	49.34		1.47	1.18–1.83			
rs874881									
CC	23.28	44.62	24.70	0.001	1		0.002	1	
CG	59.48	42.31	51.50		0.64	0.46–0.90		2.69	1.51–4.80
GG	17.24	13.08	22.90		0.43	0.27–0.68		2.53	1.15–5.58
C	53.03	65.77	50.90	0.004	1.43	1.16–1.77	0.003	0.59	0.41–0.84
G	46.95	34.23	49.10		0.69	0.56–0.86		1.70	1.18–2.45

F%: Frequency in percentage; ^++^ All individuals with RA and RA-ILD are included compared to individuals without diseases CHS. *p* * = adjusted for age, sex, tobacco index, and AIM panel. A value of *p* < 0.05 is considered significant, OR: Odds ratio, CI: 95% confidence interval.

**Table 4 cells-12-02235-t004:** Analysis of genotypes and alleles of SNV in *PADI4*. All rheumatoid arthritis and CHS groups.

*PADI4* Haplotype	Arthritis^++^ (*n* = 251, HF%)	CHS (*n* = 616, HF%)	*p*	OR (95% CI)
GTTC	41.60	35.10	0.018	1.40 (1.12–1.74)
ACCG	33.30	30.80	0.501	
GTCG	9.00	13.10	0.021	0.65 (0.44–0.94)
ACTC	7.00	10.60	0.029	0.64 (0.42–0.94)
GTCC	2.40	4.00	0.318	
ATTC	2.30	1.10	0.215	
GCCG	2.20	1.20	0.285	
ACTG	0.40	1.80	0.009	0.12 (0.02–0.87)

^++^ All individuals with RA, including those in the RA-ILD group; CHS: Clinical healthy subjects. HF% = haplotype frequency in percentage; OR: Odds ratio, 95% CI: 95% Confidence Interval. Comparison with Fisher’s exact test.

**Table 5 cells-12-02235-t005:** Analysis of genotypes and alleles of SNV in *PADI4* between patients’ groups.

*PADI4* Haplotype	RA-ILD (*n* = 118, HF%)	RA (*n* = 133, HF%)	*p*	OR (95% CI)
GTTC	39.80	47.40	0.107	
ACCG	37.70	32.20	0.222	
GTCG	5.00	9.90	0.051	
ACTC	7.20	2.90	0.038	2.64 (1.01–6.88)
ATTC	2.30	2.50	0.889	
GCCG	2.00	2.40	0.795	
GTCC	2.40	1.40	0.445	

F% = haplotype frequency in percentage; OR: Odds ratio, CI95%: 95% Confidence Interval. Comparison with Fisher’s exact test. RA-ILD: Interstitial Lung Disease–Rheumatoid Arthritis.

## Data Availability

The datasets generated for this study can be found in ClinVar accessions SCV001422427—SCV001422434.

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
