# Peer review of "Molecular Factors in PAD2 (PADI2) and PAD4 (PADI4) Are Associated with Interstitial Lung Disease Susceptibility in Rheumatoid Arthritis Patients"

_cells, 2023, doi:10.3390/cells12182235_

Round 1

Reviewer 1 Report

Overall, the study investigates an exciting and clinically relevant topic by exploring the potential association between molecular factors in PAD2 and PAD4 with interstitial lung disease (ILD) susceptibility in rheumatoid arthritis (RA) patients. The findings have the potential to impact clinical management and shed light on the underlying mechanisms of ILD development in this patient population. However, I have a few comments that I believe will further enhance the quality of the manuscript.

1. In the abstract, the ILD and BLA should be shown the full name when first mentioned in the paper, just in case some readers are unfamiliar with them.

2. The introduction provides a general overview of the topic, but it would benefit from a more comprehensive review of relevant literature on both PAD2 and PAD4 enzymes, their roles in RA pathogenesis, and their potential involvement in autoimmune-related lung diseases.

Author Response

Dear Reviewer 1

We have thoroughly reviewed each of the comments you provided. Your detailed insights and recommendations have been constructive in identifying areas where our article could be strengthened and clarified. Below, we outline how we have addressed your comments:

We have adjusted the citation format and made changes to specific sections; these modifications were carried out to enhance the visual presentation of figures and tables. It was necessary to make these adjustments as the insertion of new text impacted the placement of figures and tables across different pages. Notably, any additions or modifications, including new sentences, have been marked red to ensure clear visibility.

Reviewer comment: Overall, the study investigates an exciting and clinically relevant topic by exploring the potential association between molecular factors in PAD2 and PAD4 with interstitial lung disease (ILD) susceptibility in rheumatoid arthritis (RA) patients. The findings have the potential to impact clinical management and shed light on the underlying mechanisms of ILD development in this patient population. However, I have a few comments that I believe will further enhance the quality of the manuscript.

  1. In the abstract, the ILD and BLA should be shown the full name when first mentioned in the paper, just in case some readers are unfamiliar with them.

R: We extend our gratitude for your input. The text within the abstract has undergone modification (indicated in red in the main manuscript file).

. We aimed to identify single nucleotide variants (SNV) in PADI2 and PADI4 genes (PAD2 and PAD4 proteins, respectively) associated with susceptibility to interstitial lung disease (ILD) in RA patients and the PAD2 and PAD4 levels. Material and methods: … The ELISA technique determined PAD2 and PAD4 levels in serum and bronchoalveolar lavage (BAL) samples, and the population structure was evaluated using 14 informative ancestry markers…

  1. The introduction provides a general overview of the topic, but it would benefit from a more comprehensive review of relevant literature on both PAD2 and PAD4 enzymes, their roles in RA pathogenesis, and their potential involvement in autoimmune-related lung diseases.

R: We appreciate your observation. Articles discussing the involvement of PAD enzymes and proteins associated with the protein citrullination process in ILD have been included (indicated in red).

… In RA, specific autoantibodies have been detected as anti-citrullinated protein/peptide antibodies (ACPA), and tobacco smoking has been associated with autoimmunity and increased ACPA production [3,4]. Citrullination is a post-translational change catalyzed by peptidyl arginine deiminase (PAD) enzymes [5], modifying protein structure and increasing ACPA recognition. In the early stages of RA and idiopathic pulmonary fibrosis, soluble autoantibodies such as IgA-ACPA and soluble IgM-ACPA have been recognized, demonstrating a correlation with disease activity and smoking [6–8].

Members of the PAD family, PAD2 and PAD4, play roles in both synovial joint and lung tissues. Elevated levels of these enzymes and mRNA expression have been detected during inflammatory and fibrosis processes [9,10].…

Reviewer 2 Report

I consider this manuscript as interesting with novel data. One spelling mistake I've found in last paragraph. There is AR instead of RA. Please correct that.

Author Response

Dear Reviewer 2

Thank you for your kind review. We have thoroughly assessed each of the comments you provided. Your detailed insights and recommendations have been constructive in identifying areas where our article could be strengthened and clarified. Below, we outline how we have addressed your comments:

Reviewer comment: I consider this manuscript as interesting with novel data. One spelling mistake I've found in last paragraph. There is AR instead of RA. Please correct that.

R: We greatly appreciate your feedback. Yes, it was indeed an error that has been rectified, and it is highlighted in red within the text in the main file manuscript.

Conclusions

... The SNVs rs1748033 in PADI4, rs2057094, and rs2076615 in PADI2 were associated with RA development but not with ILD, as well as haplotype in PADI4 ACTC in patients with RA-ILD…

Finally, We have adjusted the citation format and made changes to specific sections; these modifications were carried out to enhance the visual presentation of figures and tables. It was necessary to make these adjustments as the insertion of new text impacted the placement of figures and tables across different pages. Notably, any additions or modifications, including new sentences, have been marked red to ensure clear visibility.

Reviewer 3 Report

1. This study effectively demonstrates the types of SNVs in PADI2 and PADI4 associated with RA and RA-ILD. However, there are a few points that need to be revised.

2. In the baseline characteristics, please present the data of the healthy subjects’ group, the mediation of RA-ILD and RA patients, and the racial distribution.

3. Were there any differences in the genotypes or alleles of SNVs between different races?

None

Author Response

Dear Reviewer 3

We have thoroughly reviewed each of the comments you provided. Your detailed insights and recommendations have been constructive in identifying areas where our article could be strengthened and clarified. Below, we outline how we have addressed your comments:

Reviewer comments:

  1. This study effectively demonstrates the types of SNVs in PADI2 and PADI4 associated with RA and RA-ILD. However, there are a few points that need to be revised.

R: We sincerely appreciate your insightful observations and enriching comments.

  1. In the baseline characteristics, please present the data of the healthy subjects’ group, the mediation of RA-ILD and RA patients, and the racial distribution.

R: Thank you for this critical observation. We have added the following information in the text and Table 1. Additionally, the characteristics of the CHS group, in which the proteins were determined, have been mentioned in the supplementary table, with the added information highlighted in red in the main file manuscript.

  1. Results

3.1. Also, 616 clinically healthy subjects were included; the median age was 49.24 years, 58% were men, and 42.19% were current and former smokers. Spirometry parameters indicate normal pulmonary function and exposure factors, as shown in Table 1.

  1. Materials and Methods

2.3 Section. All patients present treatment with immunosuppressive therapy: methotrexate, predni-sone, chloroquine, or hydroxychloroquine, among others (or combination). None of the patients were undergoing antifibrotic therapy at the time of sample collection.

The genetic distribution is a particular topic in the Mexican Mestizo population.

Therefore, ancestry markers mentioned in the following sections were included. In Section 2.3, [“All included individuals self-referred as Mexican-Mestizos for three generations born in México and not biologically related.”] and Section 2.7 presents the reference populations.

Silva-Zolezzi and Cols in 2009 (Reference 27) identified and described the genetic diversity, linkage disequilibrium, and haplotype sharing analyzed in Mexican Mestizos, distinctions crucial for disease association studies. Mexican Amerindian and HapMap data were used to assess ancestry. Findings highlight genetic variation, advocate efficient tag SNP selection, and mark pioneering Latin American admixture-genotyping initiative, identifying 14 private SNPs within the Mexican population.

Furthermore, in our same study group, research conducted by Pérez-Rubio G. (DOI: 10.3109/15412555.2016.1161017) assessed 251 variants employed as AIMs and demonstrated the genetic structure of the Mexican mestizo population among self-reported and unrelated individuals from the same study center.

Therefore, our analysis used these 14 private SNPs as covariates to identify whether ILD or RA was associated with changes in the PADI2 and PADI4 genes. Additionally, we examined if these SNPs were more frequently present in the native populations (ZAP) or CEU (used as reference populations).

  1. Were there any differences in the genotypes or alleles of SNVs between different races?

R: We greatly appreciate your comments. We did not conduct this specific analysis, as our focus was on the SNP in PADI2 and PADI4 and their implications for patients carrying alleles associated with ILD risk in RA patients. Nonetheless, we have considered this information as a covariate with influence in the CHS group assessed through AIMs.

Round 2

Reviewer 3 Report

The authors provided satisfactory corrections after reviewers' suggestions. The article can be accepted for publication now.